# Non-bonded force field model with advanced restrained electrostatic potential charges (RESP2)

Michael Schauperl [1], Paul S. Nerenberg [2], Hyesu Jang [3], Lee-Ping Wang [3], Christopher I. Bayly [4], David L. Mobley [5] & Michael K. Gilson [1]✉

The restrained electrostatic potential (RESP) approach is a highly regarded and widely used method of assigning partial charges to molecules for simulations. RESP uses a quantum-mechanical method that yields fortuitous overpolarization and thereby accounts only approximately for self-polarization of molecules in the condensed phase. Here we present RESP2, a next generation of this approach, where the polarity of the charges is tuned by a parameter, δ, which scales the contributions from gas- and aqueous-phase calculations. When the complete non-bonded force field model, including Lennard-Jones parameters, is optimized to liquid properties, improved accuracy is achieved, even with this reduced set of five Lennard-Jones types. We argue that RESP2 with $\delta \approx 0.6$ (60% aqueous, 40% gas-phase charges) is an accurate and robust method of generating partial charges, and that a small set of Lennard-Jones types is a good starting point for a systematic re-optimization of this important non-bonded term.

[1] Skaggs School of Pharmacy and Pharmaceutical Sciences, University of California, San Diego, CA 92093, USA. [2] Departments of Physics & Astronomy and Biological Sciences, California State University, Los Angeles, CA 90032, USA. [3] Department of Chemistry, University of California, Davis, CA 95616, USA. [4] OpenEye Scientific Software Inc., Santa Fe, NM 87508, USA. [5] Department of Pharmaceutical Sciences and Department of Chemistry, University of California, Irvine, CA 92697, USA. ✉email: mgilson@health.ucsd.edu

Molecular simulations are widely used to study chemical and biophysical processes at the atomistic level[1,2]. Applications include modeling of macromolecular interactions[3–6], protein folding[7], and drug design[8,9]. Because calculations with high-level quantum-mechanical (QM) methods are too slow for many systems of interest, simulations typically use fast, empirical potential functions known as empirical force fields (FFs)[10–12]. Instead of treating electronic degrees of freedom explicitly, FFs treat them implicitly via analytical energy terms

**Fig. 1 Mean errors in electrostatic properties.** Mean errors in molecular dipole moments **a** and electrostatic potentials **b** across 71 test compounds, relative to reference double-hybrid calculations, for various QM methods. Compute time requirements, normalized to the duration of the corresponding HF/6–31G* calculations, are provided as well **c**.

containing parameters that are empirically adjusted to replicate experimental and quantum chemistry reference data. The accuracy of simulations performed with empirical FFs thus depends critically on the accuracy of these parameters.

Non-bonded interactions, comprising dispersion, steric repulsion, and electrostatic interactions, make large contributions to atomistic forces and energies[13], so it is essential that empirical FFs treat them accurately. Dispersion forces and steric repulsion are commonly modeled by a Lennard-Jones (LJ) potential, though more complex functional forms have been proposed and used to model these interactions[14]. Electrostatic interactions include Coulombic interactions among the permanent charges of molecules, as well as interactions involving field-induced shifts in electron density, i.e., electronic polarization. Despite important advances in FFs that treat electronic polarization explicitly[15–17], fixed-charge FFs that treat electronic polarization implicitly are still widely used, as they allow more thorough conformational sampling with the same computing resources. Thus, optimization of fixed-charge FFs would immediately benefit applications that require efficient conformational sampling; it would also define a baseline of accuracy that a polarizable FF should exceed.

Common methods to generate partial atomic charges for fixed-charge FFs are either based on atoms-in-molecules approaches, e.g., the Hirshfeld or iterative stockholder methods, or are optimized to reproduce the electrostatic potential (ESP) around a molecule[18–29]. Among the most popular methods for small molecules are restrained electrostatic potential (RESP, hereafter called RESP1)[20] and AM1-BCC (refs. [21,22]). Both generate partial charges designed to reproduce the ESPs of molecules in gas phase as computed at the Hartree–Fock (HF)[30,31] level with the 6–31G* basis set[32]. This QM method fortuitously overestimates the gas-phase polarity of molecules by about the right amount to yield charges appropriate for hydrated molecules, which are polarized by the reaction field of the solvent[20,33,34]. An interesting nuance of prepolarizing partial charges for use in nonpolarizable FFs is that, since the energetic cost of polarization is neglected, it is desirable to underestimate the amount of true polarization so that the energetic stabilization of favorable polar interactions in the FF is not overestimated[35]. However, the overpolarization of HF/6–31G* still appears to underestimate the polarization typically induced by hydration and to be inconsistent across different molecules[36]. It is therefore of interest to consider whether ESPs computed with higher-level QM methods could provide more accurate charges and thus more accurate simulations.

Prior studies have explored this idea. For example, Cerutti and coworkers developed the implicitly polarized charge method (IPolQ), in the context of an AMBER biomolecular FFs (ref. [37]). In IPolQ, partial charges are obtained by performing two MP2/cc-pV(T + d)Z QM calculations, one in the gas phase and the other in an explicit solvent reaction field derived from MM simulation snapshots, then averaging the two sets of fitted charges[19]. Muddana and coworkers subsequently suggested the IPolQ-Mod approach, which is identical in spirit to IPolQ but saves time by running QM in the context of an implicit hydration model, rather than using multiple simulation snapshots[38]. Following the physically motivated logic of Karamertzanis and coworkers[39], IPolQ methods weight the gas-phase and aqueous-phase charges equally in arriving at the final charge set. In a similar spirit, Duan and coworkers fitted amino acid charges to ESPs computed with B3LYP/cc-pVTZ//HF/6-31G** for molecules immersed in a dielectric continuum with a dielectric constant of 4 (ref. [18]). This choice of dielectric constant was intended to mimic the interior of a protein, and yields charges intermediate between those of the gas and aqueous phases.

Regardless of what improvement is brought to the calculation of partial atomic charges, the total non-bonded energy depends

not only on charges but also on LJ interactions, so using more accurate charges without adjusting LJ parameters also may not in fact afford greater accuracy. Indeed, both Cerutti and coworkers[19], and Mobley and coworkers[40] found that changing the charge parameters alone did not increase the accuracy of simulations. In addition, although Karamertzanis and coworkers[39] argued that aqueous-phase charges should be a 50:50 average of QM charges computed in the gas and aqueous phases, this balance might not actually lead to optimal accuracy, because the simplified functional forms used in most FFs may require some cancellation of error among terms to reach greatest accuracy. Nonetheless, we are not aware of any systematic study of whether ESP charges derived as mixtures of gas- and aqueous-phase charges can yield greater simulation accuracy in the context of co-optimized LJ parameters, or of how simulation accuracy depends on the mixing weights placed on gas- vs aqueous-phase charges.

Here we address these issues. We determine a suitable level of QM theory for generation of accurate ESPs and then compute partial atomic charges as linear combinations of gas- and aqueous-phase charges, with mixing parameter $\delta$. Increases in accuracy relative to RESP1 may derive both from the use of QM calculations more accurate than HF/6–31G*, and from the fine-tuning of the mixing parameter $\delta$ against experimental condensed-phase data. This approach, termed RESP2, decouples the calculation of ESP charges from reliance on the arbitrary and inconsistent pattern of overpolarization afforded by HF/6–31G* calculations. We evaluate RESP2 by using the ForceBalance software[41] to co-optimize the mixing parameter $\delta$ and LJ parameters against experimentally measured properties of pure organic liquids. The resulting parameter sets are then tested against a second set of experimental data, including added data types expected to be sensitive to partial atomic charges. The results are compared with those obtained with standard RESP1, both with LJ parameters drawn from an existing FF and with optimized LJ parameters. Implications for further development and for fixed point-charge FFs in general, as well as further directions are given.

## Results

### QM methods for ESP calculations.
We tested QM methods to assess their computational speed and the accuracy of the gas-phase dipole moments and ESPs they afford for 71 test compounds, where accuracy was assessed based on comparisons with higher-level calculations using DSD-PBEP86-D3BJ (ref. [42]) with an aug-cc-pV(Q + d)Z basis set[43], as detailed in the Methods section. As shown in Fig. 1, all post-HF methods yield more accurate dipole moments and ESPs than HF. Although the choice of functional does not seem to be very critical, the usage of either partially (jun) or fully (aug) augmented basis sets, improves accuracy for these electrostatic properties. This has been observed before[44,45], though rarely in the fixed-charge FF literature. The computational cost varies widely across basis sets and functionals. We selected PW6B95/aug-cc-pV(D + d)Z as a solid combination of speed and accuracy for use with RESP2. With the current version of psi4 (ref. [46]; v.1.3.2) these gas-phase calculations are around seven times slower than HF/6–31G*, which is historically the de facto standard for RESP1. When used with implicit solvent, the calculations are 20 times slower than HF/6–31G*. Carrying out a full RESP2 calculation for one of the present training or test set compounds takes ~30 min with psi4 on a single CPU. It is worth noting that, because RESP2 does not rely on the fortuitous overpolarization afforded by HF/6–31G* but is instead based on calculations that aim for maximum accuracy, PW6B95/aug-cc-pV(D + d)Z could appropriately be replaced by other methods that also yield good agreement with gold standard reference methods.

### Assessment of charge models with baseline LJ parameters.
We first compared the accuracy of liquid state properties and hydration free energies (HFE) computed using RESP1 and RESP2 with values of $\delta$ ranging from 0 to 1, both in combination with existing LJ and valence parameters from the SMIRNOFF99Frosst v.1.0.7 FF. For brevity, we will use the notation RESP2$_\delta$ so, for example, RESP2 with $\delta = 0.6$ is called RESP2$_{0.6}$. Although no LJ parameters were trained at this stage, we report the training and test set results separately, to facilitate comparison with the corresponding results following optimization of the LJ parameters (section 3.3). Densities and heats of vaporization (HOV) of pure organic liquids computed using RESP2 charges with $\delta$ values near 0.5 are about as accurate as those with RESP1 charges, based on both mean unsigned errors (MUE; Fig. 2) and mean signed errors (MSE; Supplementary Fig. 4). However, pure liquid dielectric constants are somewhat more accurate with RESP2 charges when $\delta > 0.2$. The pattern of changes in accuracy as $\delta$ moves away from 0.5 differ across properties, with HOV and densities somewhat more accurate overall for $\delta > 0.5$ but dielectric constants and HFE more accurate for $\delta > 0.5$.

### Assessment of charge models with optimized LJ parameters.
The accuracy of non-bonded interactions is controlled by the choice of both partial charges and LJ parameters. As a consequence, the utility of a given charge set cannot be properly assessed unless LJ parameters are adjusted along with it. We therefore examined the accuracy (MUE) afforded by RESP1 (Fig. 3, green line) and by RESP2 charges with a range of $\delta$ values (Fig. 3, blue line), when LJ parameters are optimized separately for each charge assignment method based on experimental densities and HOV (MSE, Supplementary Fig. 5). In order to enable efficient optimization, we restricted the number of LJ types to five, corresponding to elements, C, O, and N, along with polar and nonpolar H, for a total of ten LJ parameters, $\epsilon$ and $r_{min-half}$ for each type. (Using a single H type led to markedly worse agreement with experiment; see sample results in Supplementary Table 4). The results obtained by optimization of this parsimonious set of LJ parameters are furthermore compared with results obtained using RESP1 and the full set of standard (non-optimized) SMIRNOFF99Frosst v1.0.7 LJ parameters (Fig. 3, red line). The combinations of RESP1 and RESP2 with optimized LJ parameters are referred to as RESP1/LJ opt and RESP2/ LJ opt, respectively, while RESP1 with baseline LJ parameters is termed RESP1/SMIRNOFF. The optimized LJ parameters of selected LJ models can be found in Supplementary Fig. 6 and Supplementary Notes 2–4.

One broad observation is that, when LJ parameters are adjusted for each value of $\delta$ in the RESP2 model, the level of error becomes much less sensitive to $\delta$, so the blue curves in Fig. 3 are much flatter than those in Fig. 2. This reflects the strong interdependence of the LJ parameters with the charge model, wherein adjustment of LJ parameters can allow a range of charge models to yield similar levels of accuracy. Considering the results in more detail, one may see that, for the training set data (Fig. 3a, b), RESP1/LJ opt (green) and RESP2/LJ opt (blue) afford consistently lower errors than RESP1/SMIRNOFF (red). For the test set densities (Fig. 3c), RESP1/LJ opt and RESP2/LJ opt with all values of $\delta$ give consistently lower error than RESP1/SMIRNOFF. Interestingly, RESP2/LJ opt with $\delta < 0.8$ yields HOV similar to RESP1/SMIRNOFF, while RESP1/LJ opt affords somewhat greater accuracy. However, RESP2/LJ opt with $\delta < 0.2$ yields the most accurate dielectric constants, as perhaps expected given that this property may be particularly sensitive to the quality of the charge model. HFE computed with RESP2/LJ opt are more accurate than those computed with RESP1/LJ opt, except when

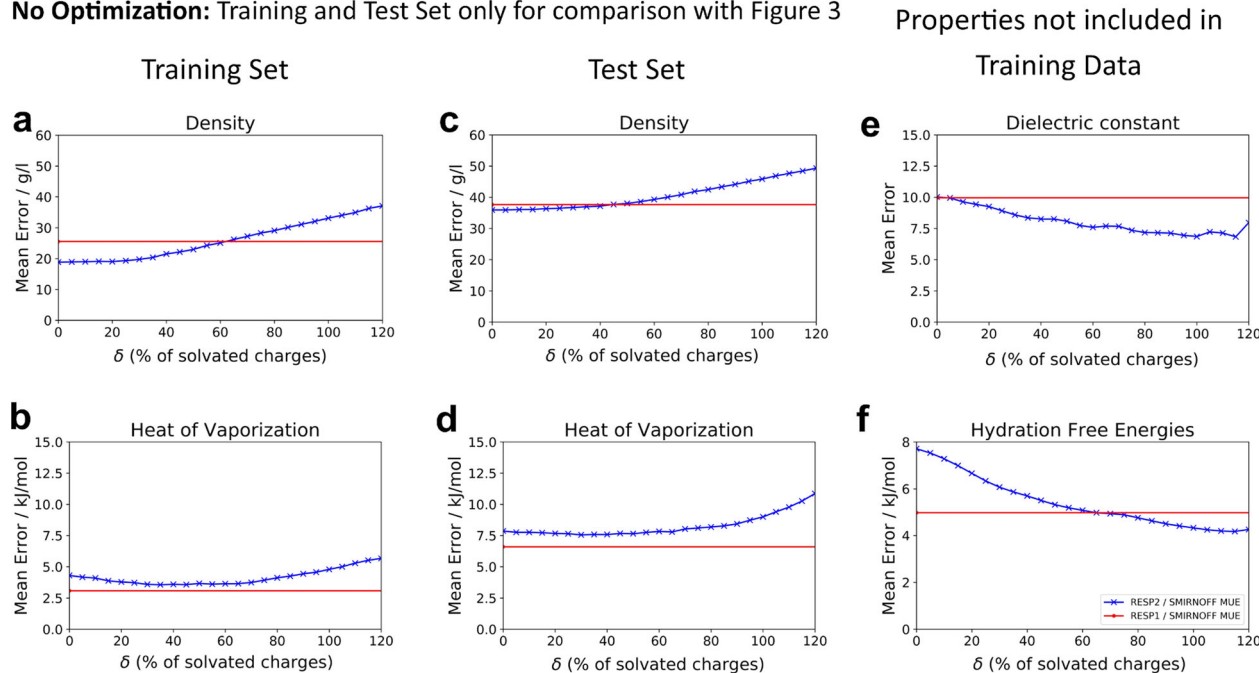

**Fig. 2 MUE with SMIRNOFF LJ parameters.** Comparison of theoretical and experimental results (MUE) as a function of the charge mixing parameter $\delta$ with SMIRNOFF LJ parameters. No parameters are optimized for these results. Separation of training and test set is kept to facilitate comparison with Fig. 3. Mean error for densities and HOV for the training set **a**, **b** and test set **c**, **d**. Mean error for the dielectric constants **e** of the test set, and HFE error for all molecules in the FreeSolv database and either in the test or training set **f**. The red line are results obtained with RESP1 charges and is used as a reference.

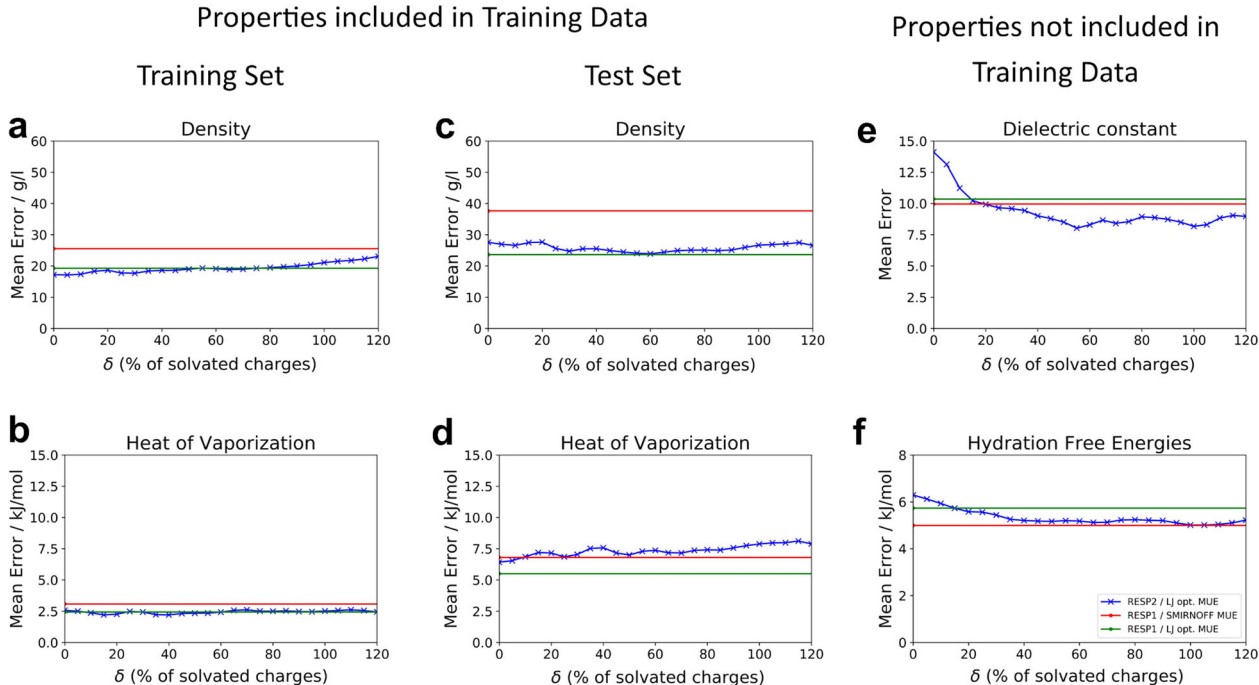

**Fig. 3 MUE with reoptimized LJ parameters.** Comparison of theoretical and experimental results as a function of the RESP2 charge mixing parameter $\delta$ with reoptimized LJ parameters. Mean error for densities and HOV for the training set **a**, **b** and test set **c**, **d**. Mean error for the dielectric constants **e** for the test set, and HFE error for all molecules in the FreeSolv database and either in the test or training set **f**. The red lines are results obtained with RESP1 charges and smirnoff99Frosstv1.0.7 LJ parameters. The green lines are results with RESP1 charges and reoptimized LJ parameters.

the most gas phase-like ($\delta < 0.2$) RESP2 charges are used. HFE computed with RESP2/LJ opt are similar in accuracy to those obtained with RESP1/SMIRNOFF. Note, however, that the HFE calculations are not pure tests of the present FF approaches, because they use the TIP3P water model, and other water models

will give different results. Overall, RESP2/LJ opt with $\delta < 0.6$ led to improved dielectric constants, densities, and HFE, but some loss in accuracy for HOV, relative to RESP1/LJ opt.

To examine how overall accuracy varies with $\delta$, we graphed the average unsigned error, given relative to RESP1/SMIRNOFF, as a

function of this mixing parameter (Fig. 4). Some of the roughness of the graph appears to be noise, such as from slight variations in the convergence behavior of the ForceBalance runs with different values of $\delta$. With this in mind, the best value of $\delta$ appears to be between 0.5 and 0.7, and provisionally choose the center of this range, 0.6, as the best current choice. Note, however, that the shape of the graph depends on the weighting of the four experimental properties. Here, they have been accorded equal weight. If densities and HOV were weighted more heavily, the minimum would shift to the left. If dielectric constants and HFE were weighted more heavily, the minimum would shift to the right.

It is also of interest that optimization of even a very small set of LJ parameters can lead to markedly improved accuracy relative to the larger baseline set of LJ parameters. For most LJ types, the observed changes in $\varepsilon$ and $r_{min-half}$ relative to the initial SMIRNOFF values are subtle. The greatest change is observed for the nitrile carbon $\varepsilon$ value, which decreases from 0.21 kcal/mol to 0.085 kcal/mol. Nonetheless, molecules containing this atom type do not show a higher than average error (Supplementary Fig. 1). These observations raise the question whether all of the LJ types in smirnoff99Frosst are necessary. This issue is beyond the scope of the present study and will be addressed in a subsequent work.

**Molecular dipole moments and atomic partial charges**. Because RESP2 is based on a higher level of theory than RESP1, we

conjectured that molecular dipole moments computed with RESP2 charges would correlate better with dipole moments obtained directly from QM calculations. This expectation holds true, as shown in Fig. 5. Thus, dipole moments computed with $RESP2_{0.6}$ charges have $R^2$ values of 0.99 against both gas-phase and aqueous QM dipole moments, while RESP1 yields $R^2$ values of 0.97 and 0.96, respectively. Similar results are obtained for $RESP2_{0.5}$ (Supplementary Fig. 2). Moreover, some RESP1 dipole moments are less than the corresponding gas-phase QM results, indicating that HF/6–31G* does not consistently yield the over-polarization assumed to make RESP1 partial charges suitable for aqueous-phase simulations[36]. In contrast, RESP2 dipole moments are never below the corresponding QM gas-phase results. Interestingly, RESP2 charges tend to yield 10% larger dipole moments than RESP1, as evident by inspection of Fig. 5 and from the regression slopes provided there. The differences between individual dipole moments obtained from RESP1 and $RESP2_{0.6}$ charges range up to 30% (Supplementary Table 1).

It is also of interest to compare RESP1 and $RESP2_{0.6}$ partial charges directly. As shown in Fig. 6, the differences are on par with those between QM charges computed for gas vs aqueous phase, but less than the differences between RESP1 and widely used AM1-BCC charges. Although the molecular dipoles moments are increased by ~10% (above), the charges are of a similar magnitude for $RESP2_{0.6}$ and RESP1, as indicated by the regression coefficient (slope) of 0.999.

## Discussion

The present study defines and tests a non-bonded interaction model based on RESP2 charges, a logical extension of prior work aimed at developing physically meaningful, atom-centered, fixed, partial charges for FFs used in molecular simulations. This work seeks to overcome limitations in accuracy of RESP1 that result from its reliance on the HF/6–31G* QM method and spotlights the importance of allowing LJ parameters to be adjusted along with a partial charge model, in order to generate an optimal representation of non-bonded interactions.

Gas-phase HF/6–31G* calculations are widely used to obtain ESP-based partial charges in the RESP1 method[20]. Although this method/basis set combination leads to overpolarization and thus makes the resulting charges plausible for the condensed phase, the degree of overpolarization is inconsistent across compounds[36]. For example, as reported in the Results section, RESP1 partial charges sometimes lead to molecular dipole moments even smaller than reference gas-phase QM dipole moments. Moreover, Duan and coworkers have argued that the pattern of

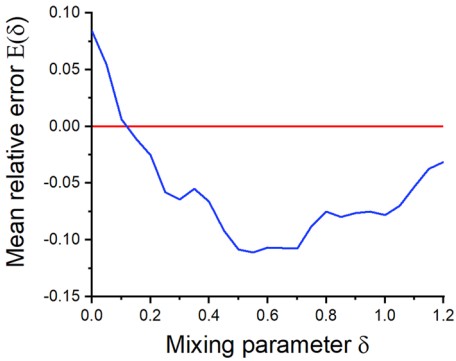

**Fig. 4 Average error as function of the mixing parameter $\delta$.** Average unsigned error, relative to baseline RESP1/SMIRNOFF (red), of test set predictions as a function of the RESP2 charge mixing parameter $\delta$ with reoptimized LJ parameters (blue).

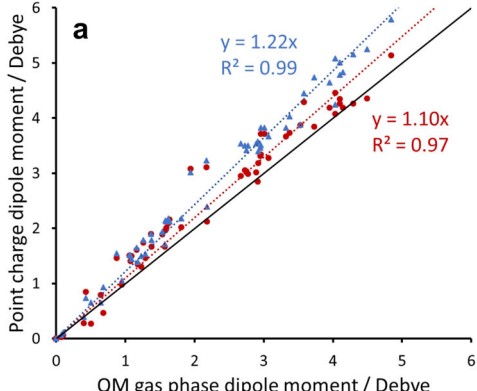
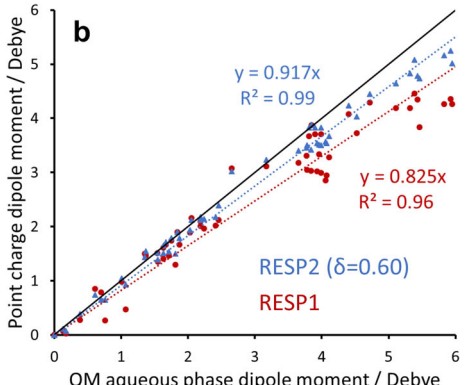

**Fig. 5 Comparison of molecular dipole moments from partial charges with those obtained directly from QM calculations.** Scatter plots of molecular dipole moments based on RESP1 and $RESP2_{0.6}$ point charges against molecular dipole moments based on electron density from QM (PW6B95/aug-cc-pV (D + d)Z) calculations in gas phase **a** and aqueous phase **b**. Black line: slope of unity.

overpolarization by HF/6–31G* does not match the actual polarization induced by the reaction field of a solvent, as the former will tend to overpolarize all parts of a molecule, while the latter will tend to polarize mainly solvent-exposed parts[18].

The RESP2 approach is similar in spirit to previous approaches aimed at overcoming these issues, including that of Duan and coworkers, who used higher-level QM methods with an implicit solvent having a dielectric constant of 4 to obtain high-quality partial charges suitable for the condensed phase; the IPolQ method, which obtains partial charges by a 0.5 scaling between higher-level gas-phase QM calculations and QM calculations carried out in an ensemble of explicit water conformations drawn from simulations; and to the IPolQ-Mod method, which may be viewed as $RESP2_{0.5}$ run with a different but still higher-level QM method. However, the present study is distinguished from prior work by our assigning the scaling between gas- and condensed-phase charges to the adjustable parameter, $\delta$, whose value is adjusted based on empirical fitting to experimental data. The empirical fitting enables to compensate for possible errors and bias introduced by other parts of the charge derivation, such as the details of the PCM model, and for limitations of the FF's functional form. This approach has the additional benefit that no additional QM calculations are needed to adjust the polarity of the charge model when fitting against additional experimental data or adjusting charges in the context of a modified functional form; one only needs to adjust the value of $\delta$.

It is perhaps worth commenting on our use of the high dielectric constant of water in the condensed-phase QM calculations used in RESP2. This choice might be expected to lead to excessive polarization of the compounds making up organic liquids with much lower dielectric constants. The fact that reasonably good results are obtained despite this simplification may result from the fact that apolar molecules generate only a weak reaction field no matter what the surrounding dielectric constant, so the precise value of the dielectric constant used in computing their charges may not matter much. For polar molecules, on the other hand, where the choice of dielectric constant in the calculations has a greater effect on the fitted partial charges, the liquids have a dielectric constant closer to water, making this choice more reasonable.

We have not only tested RESP2 in the context of existing LJ parameters, but have also optimized the entire non-bonded part of the FF by adjusting LJ parameters in the context of RESP1 charges and of RESP2 charges with a range of scaling parameters, in order to learn which method is capable of giving best agreement with experiment. A key observation is that optimizing LJ parameters significantly reduces the errors relative to experiment for all of the charge sets. These improvements are in spite of the small number of LJ types used in our work, which assigns the same parameters to all instances of each element, except for a split of hydrogen into polar and nonpolar types. Further, when LJ parameters are optimized for each proposed charge model, the sensitivity to the choice of charge model decreases considerably. Thus, the empirical utility of a charge model cannot be fully assessed without determining how well it performs with correspondingly optimized LJ parameters. Indeed, we would argue that a charge model should ultimately be assessed within the context of a full optimization of the entire FF, based on a range of suitable reference data that can include both experimental and QM results; this is a central strategy of the Open FF project[47]. The present LJ optimization results also suggest that the many LJ types in most current FFs may not be needed, because similarly accurate results might be obtainable with a much smaller number of types. Automated optimization methods, such as the ForceBalance tool used here[41], open the possibility of systematically addressing this issue in future work.

When RESP2 was tested along with baseline SMIRNOFF99-Frosst parameters (RESP2/SMIRNOFF), the accuracy of densities

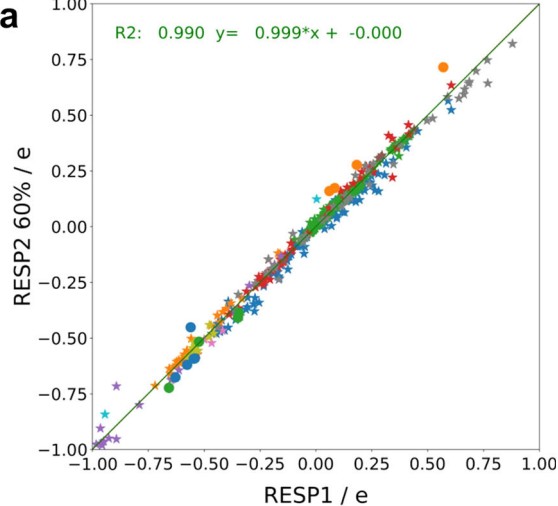

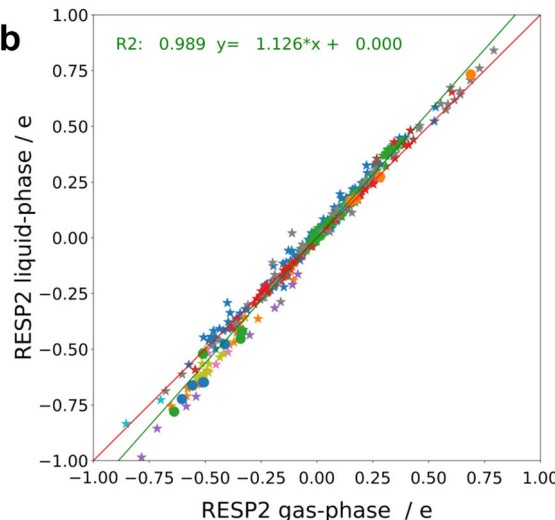

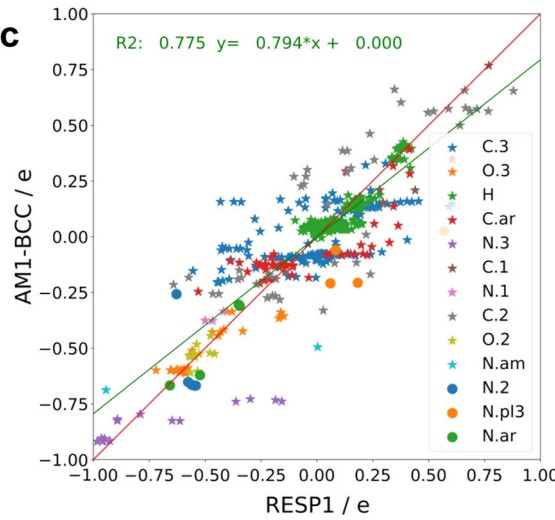

**Fig. 6 Comparison of partial atomic charges generated by various methods.** Charge comparisons between RESP1 charges and RESP2 charges with a mixing parameter of 0.6 **a**; RESP2 gas phase and RESP2 implicit solvent charges **b**, and AM1-BCC and RESP1 charges **c**.

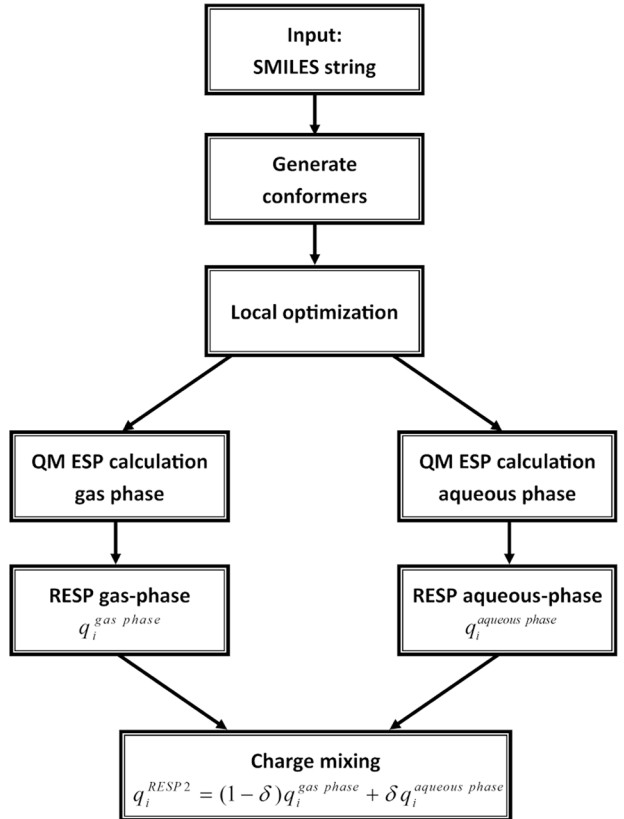

**Fig. 7 Flow chart to generate RESP2 charges from SMILES strings.** Conformers are generated using Openeye's Omega. QM calculations are done with psi4. Respyte is used for the ESP point selection and the charge fitting stage.

was well-preserved, but HOV became somewhat less accurate, even with optimal values of ~0.5–0.6 for $\delta$. On the other hand, dielectric constants immediately become more accurate with RESP2 charges, presumably because these are particularly sensitive to the fidelity of the charge distribution afforded by the charge model. Also, as noted in the Results section, molecular dipole moments computed with RESP2 correlate better with QM results than dipole moments from RESP1 charges. Interestingly, the overall accuracy of the liquid state properties afforded by all charge models improved when LJ parameters were optimized. This was true even for RESP1, even though one might have expected that "standard" LJ parameters would have been optimized over time for use with the long-standing RESP1 method. These results are consistent with the findings of Cerutti et al., who found it necessary to begin adjusting LJ parameters to improve results with the IPolQ charge model. Note, however, that they did not check how adjusting LJ parameters might also improve accuracy with the baseline RESP1 charge model[19].

The present analysis supports the use of RESP2 with $\delta \approx 0.6$ (i.e., RESP2$_{0.6}$), with LJ parameters optimized along with RESP2$_{0.6}$ charges, as this show the biggest improvement for reproducing experimental properties in comparison to RESP1/ SMIRNOFF (all properties are weighted equally). Note that the scaling factor of 0.6 is close to the value of 0.5 used with IPolQ (ref. [19]) and IPolQ-Mod (ref. [38]) based on physical reasoning[39]. In comparison with RESP1/LJ opt, RESP2$_{0.6}$/LJ opt gives balanced results across densities, HOV, and HFE, along with improved dielectric constants. However, this initial study cannot definitively establish RESP2 as superior to RESP1. Additional studies spanning a range of experimental observables and against the

background of additional FF optimizations will be needed to fully compare the two methods. However, we anticipate that the accuracy of current fixed-charge FFs can be improved by moving to higher-level QM calculations, as done here. An additional advantage of the present method is that the scaling parameter $\delta$ is a simple, physically motivated control to tune the overall polarity of the charges generated. This may be co-optimized with the LJ parameters, allowing for straightforward tuning of charges along with the LJ parameters. It is also worth noting that optimization of valence terms, especially torsions, along with the current non-bonded forces should result in further improvement in accuracy. Ultimately, it will be important to assess the accuracy of FFs based on the present approach with that of other popular force fields, e.g. OPLS3, CHARMM, and GAFF2 (refs. [48,49]), which provide highly competitive performance relative to RESP1/GAFF, which in turn is similar to RESP1/SMIRNOFF (refs. [48,50]).

In considering the present results, it is also worth keeping in mind that, although the HFE calculations are a useful reality check, they do not cleanly test the charge models. This is because they rely on a specific water model, TIP3P, which was chosen somewhat arbitrarily from among other excellent options, such as TIP3P-FB (ref. [41]), SPC/E (refs. [51,52]), TIP4P-Ew (ref. [53]), and OPC (ref. [54]). We anticipate that changing to a different water model will have a nonuniform effect on the accuracy of the various models tested here. It is also worth noting that both HFE and HOV could place a needless burden on fixed-charge models whose intended use is only for the calculation of condensed-phase properties. This is because the changes in electronic polarization that are not explicitly treated by fixed-charge models are greatest in the setting of a gas-to-condensed-phase transfer, but a condensed-phase simulation does not need to handle such scenarios. By the same token, FFs that do not account explicitly for electronic polarization are not able to model the electronic contribution to the dielectric constants of organic liquids. As a consequence, they tend to underestimate these dielectric constants, particularly for nonpolar compounds where orientational polarizability cannot compensate for the lack of electronic polarization.

Thus, although dielectric constants, HOV, and HFE have long been a mainstay for the adjustment and testing of FFs, it would seem preferable to focus in the future on other experimental properties that involve only condensed-phase processes and can be modeled accurately with non-polarizable FFs e.g., liquid mixture data and surface tension[55].

The chief drawback of RESP2 is that it is slower than RESP1, because it requires two higher-level QM calculations, one of them with an implicit solvent model, for each molecular conformer included in the calculation. This is not likely to be problematic for applications focusing on, e.g., tens of compounds, but can be burdensome for larger-scale studies. It may thus be of interest to develop what amounts to a second-generation AM1-BCC method[21,22], trained to match RESP2 instead of RESP1 charges. We envision replacing AM1 with a higher level, but still efficient QM method, and then training a new set of BCCs to agree with RESP2 charges and/or to yield accurate condensed-phase properties when used in simulations. Alternatively, fast, machine-learning (ML) methods for generating partial charges have recently been devised[56,57], and these methods need to be trained against some kind of data. The encouraging results for RESP2$_{0.6}$ found here suggest it as a promising physics-based charge model to train fast ML methods.

## Methods
**Definition and calculation of RESP2 charges**. We propose RESP2, a method of generating atom-centered partial charges for small organic molecules. To compute RESP2 charges, one carries out two separate RESP calculations, one for the

molecule of interest in gas phase, and the other for the same molecule in water; i.e., using an implicit solvent model with dielectric constant 78.39. As detailed below, each RESP calculation may use multiple conformers of the molecule. The two resulting charge sets, termed, $q_i^{\text{gas}}$ and $q_i^{\text{aqueous}}$, respectively, for atoms indexed by $i$, are then combined with a mixing parameter $\delta$.

$$q_i^{\text{RESP2}} = (1-\delta)q_i^{\text{gas}} + \delta q_i^{\text{aqueous}} \qquad (1)$$

The value of $\delta$ effectively defines the polarity of the RESP2 charges, with $\delta = 0$ providing less polar gas-phase charges and $\delta = 1$ providing more polar aqueous-phase charges. This scaling procedure has the merit of preserving the total charge of the molecule, so it can be used with both neutral and charged compounds. It also allows the polarity of the charge model to be varied without any requirement for additional QM calculations, unlike the alternative approach of running QM in implicit solvent with various values of the dielectric constant. Note, too, that setting $\delta = 0.5$ causes RESP2 charges to equal IPolQ-Mod charges[38], assuming the same QM method and implicit solvent model are used. Both methods are in the spirit of IPolQ (refs. [19,39]) with the difference that IPolQ uses explicit solvent simulations to generate the reaction field. In the present study, we systematically examine the accuracy of condensed-phase simulations carried out with a range of $\delta$ values.

The generation of RESP2 charges for the molecules in this study follows the flow chart in Fig. 7. First, up to five conformers (using the `maxconfs` keyword) were generated with the program Omega[58,59]; more conformers may be appropriate for larger and more flexible compounds. Conformers with energy >10 kcal/mol (based on the Omega energy function) above the most stable conformer were discarded. Conformationally distinct low-energy structures were selected using Omega keywords `RangeIncrement 2` and `RMSRange 0.5 1.0 1.5 2.0`. Each conformer was energy-minimized in the program psi4 (ref. [46]) with QM at the selected level of theory, PW6B95/cc-pV(D + d)Z (see the Results section). Based on visual inspection, the conformations generated by this protocol were not particularly compact and did not have a high degree of intramolecular hydrogen bonding, so they seem reasonably representative of the condensed phase. The program respyte[60] was used to generate ESP points on Merz-Singh-Kollman (MSK) shells[28] with inner and outer radii of 1.6 $R_i$ and 2.0 $R_i$, and 0.2 $R_i$ spacing between point layers where $R_i$ is the van der Waals radius (Bondi radii), and with a density of 2.4 points/$\text{Å}^2$ in each layer. The ESPs at these points were calculated with psi4 using PW6B95/aug-cc-pV (D + d)Z, first in vacuo and then with implicit solvent ($\varepsilon = 78.39$, CPCM (refs. [61,62]), Bondi radii). Like the initial RESP method, RESP2 uses a two-stage fitting protocol. In the first optimization step, all charges are allowed to change independently, with a weak hyperbolic restraint constant of 0.005 $e/a_0^2$ centered at 0.0. In the second step, chemical symmetry is enforced and only apolar parts of the molecules are refitted, with a higher restraint constant of 0.01 $e/a_0^2$. The restraints reduce the conformational dependency of the charges and to ensure chemically sensible charges on buried atoms, which otherwise might be not well defined, as previously detailed[20]. The RESP method was used to simultaneously fit a single set of partial charges $q_i^{\text{gas}}$ to the gas-phase ESPs of all conformers, with all conformations assigned equal weight. Likewise, a single set of partial charges $q_i^{\text{aqueous}}$ was fit to the aqueous-phase ESPs of all conformers. The partial charges $q_i^{\text{gas}}$ and $q_i^{\text{aqueous}}$ were then used in Eq. (1), with any desired value of $\delta$.

**Selection of QM method for RESP2.** We evaluated a number of QM methods in order to arrive at a level of theory for RESP2 that affords good accuracy at modest computational cost. Because atomic charges are not physical observables, one cannot directly assess the accuracy of the charges themselves. Therefore, we instead examined molecular dipole moments and ESPs, which are closely related to partial charges. We used a set of 71 molecules (Supplementary Fig. 3), including those previously used by Hickey and Rowley to benchmark QM calculations of electrostatic properties[44]. For all property calculations, we used molecule geometries built with Open Babel 2.4.1 (ref. [63]) and optimized with B3LYP (ref. [64])/ aug-cc-pV (Q + d)Z (refs. [65–67]). High-quality reference calculations were carried out using DSD-PBEP86-D3BJ (ref. [42]) with an aug-cc-pV(Q + d)Z basis set[43]. This method is a close relative to DSD-PBEPBE-D3BJ, which offers accuracy, for gas-phase dipoles and polarizabilities, on par with CCSD (ref. [68]) calculations at much lower computational cost[69,70]. Note that Dunning basis sets with additional tight d functions for second row atoms are necessary to reproduce molecular properties for sulfur-containing molecules with high accuracy[71]. Gaussian16 (ref. [72]) was used to select the QM method for RESP2, as double-hybrid functional properties were not available in psi4 at the time of this study. The open-source Psi4 package was used for the rest of the project. We evaluated five methods with five different basis sets against the DSD-PBEP86-D3BJ reference: the methods used were MP2 (ref. [73]), HF (refs. [30,31]), B3LYP (ref. [64]), PBE0 (ref. [74]), and PW6B95 (ref. [75]) and the bases were aug-cc-pV(D + d)Z, cc-pV(T + d)Z, jun-cc-pV(T + d)Z, aug-cc-pV(T + d)Z, and aug-cc-pV(Q + d)Z (refs. [65–67]). ESP points were selected based on (MSK) grids[28] with 17 points per unit area and ten layers (gaussian keywords `IOP(6/41=10, 6/42=17)`). Timings were noted and the dipole moments and ESPs were compared with the corresponding reference results. A method/basis combination that offered a good compromise between performance and cost was chosen for charge derivation (PW6B95/aug-cc-pV(D + d)Z; see the Results section).

**Evaluation of RESP2 charges with and without LJ parameter optimization.** We evaluated RESP2 charges for their ability to replicate experimental observables, such as the densities and HOV of pure organic liquids. The results were compared with matched evaluations of RESP1 charges. We first ran tests of these charge models in the context of otherwise unchanged SMIRNOFF v1.0.7 FF parameters[50]. Then, recognizing that the accuracy afforded by a charge model depends on the LJ parameters used with it, we examined the accuracy achievable by each charge model with LJ parameters optimized in the context of that charge model. This was done by optimizing LJ parameters against training set experimental data for a given charge set and testing the resulting partial charge/LJ combinations against a separate set of experimental data. Details of these procedures follow.

The program ForceBalance[41] was used to optimize the LJ parameters for RESP1 charges and for RESP2 with values of $\delta$ from 0 to 1 in steps of 0.05. To simplify and speed the optimizations, we limited the number of different LJ types to five: C, N, and O, polar H, and apolar H. Polar hydrogens were defined by the following extended SMARTS pattern: [#1:1]-[#7,#8]. Because each LJ type has two parameters, $r_{\text{min-half}}$ and $\varepsilon$, the optimizations were done in a ten-dimensional parameter space. Starting parameters were drawn from SMIRNOFF v1.0.7 (Supplementary Notes 1). Training was based on measured HOV and pure liquid densities of 15 molecules (Fig. 8a) with a variety of functional groups. The ForceBalance procedure was terminated when the step size for the mathematical parameters fell <0.01 or the objective function changed <1.0 between two iterations; further details are provided below in this section. The resulting parameters were tested against measured HOVs and densities for a separate set of 53 molecules (Fig. 8b), as well as the measured dielectric constants and HFE of a subset of these compounds. All experimental values for HOV were taken from ThermoML[76]. Densities were taken from ThermoML when available, and otherwise from PubChem[77]. Dielectric constants were taken from multiple sources[78]. HFE was taken from the FreeSolv database[79]. All values are summarized in Supplementary Tables 2 and 3.

The objective function used in the ForceBalance calculations is now described; further details are available elsewhere[15]. The $N$ physical parameters $\mathbf{K} = (K_1, K_2, \ldots K_N)$—here the values of $\epsilon$ and $r_{\text{min-half}}$ for each of the five LJ types —are mapped to mathematical parameters $\mathbf{k} = (k_1, k_2, \ldots k_N)$ by shifting and scaling according to the following expression

$$k_i = \frac{1}{t_i}\left(K_i - K_i^o\right) \qquad (2)$$

where $K_i^0$ is the initial value of the FF parameter $K_i$, and $1/t_i$ is a scaling factor determined by the prior for each FF parameter. Prior widths were set to 0.4184 kJ/ mol for $\varepsilon$ and 1.0 Å for $r_{\text{min-half}}$. For a training set with $M$ molecules, the objective function $L(\mathbf{k})$ contains a contribution $L_m(\mathbf{k})$ from each training set molecule $m$, which quantifies the deviation of its $P$ computed properties from experiment; and a Tikhonov regularization term, weighted by $w_{\text{reg}} = 10$, which avoids large deviations from the starting values:

$$L(\mathbf{k}) = \sum_{m=1}^{M} L_m(\mathbf{k}) + w_{\text{reg}}|\mathbf{k}|^2 \qquad (3)$$

$$L_m(\mathbf{k}) = \sum_{p=1}^{P} L_p^m(\mathbf{k}) \qquad (4)$$

$$L_p^m(\mathbf{k}) = \frac{1}{d_p^2}\left|y_p^m(\mathbf{k}) - y_{p,\text{ref}}^m\right|^2 \qquad (5)$$

Here $y_p^m(\mathbf{k})$ is the value of the $p^{th}$ property for molecule $m$ (e.g., its HOV) computed for mathematical parameters $\mathbf{k}$, and $y_{p,\text{ref}}^m$ is the experimental reference value of this property. The scaling factors $d_p^m$ balance the weighting of the properties and remove their units; we used $d_{\text{HOV}} = 0.3$ kJ/mol and $d_{\text{density}} = 30$ kg/m$^3$ for all molecules $m$.

Finally, we evaluated the overall accuracy of RESP2$_\delta$/LJ opt as a function of the mixing parameter, $\delta$, in terms of the mean relative error it affords, reported relative to the baseline RESP1/SMIRNOFF model:

$$E(\delta) = \frac{1}{N_P}\sum_{j=1}^{N_P} \frac{E_j^{\text{RESP2}_\delta/\text{LJopt}} - E_j^{\text{RESP1,SMIRNOFF}}}{E_j^{\text{RESP1,SMIRNOFF}}} \qquad (6)$$

Here $j$ indexes the $N_P = 4$ experimental properties (densities, HOV, dielectric constants, and HFE), and the MUE for the superscripted model and property $j$ is given by:

$$E_j^{\text{model}} = \frac{1}{N_d}\sum_{i=1}^{N_d}\left|x_{ij}^{\text{model}} - x_{ij}^{\text{expt}}\right| \qquad (7)$$

where $N_d$ is the number of test set data, $x_{ij}^{\text{model}}$ is the value of property $j$ for molecule $i$, computed with either the RESP2$_\delta$/LJ opt or RESP1/SMIRNOFF model, and $x_i^{\text{expt}}$ is the corresponding experimental result.

**Simulation details.** In the course of its iterative parameter optimization, ForceBalance called OpenMM[80] to compute physical properties from molecular simulations. In each iteration, a gas-phase and a liquid-phase simulation at

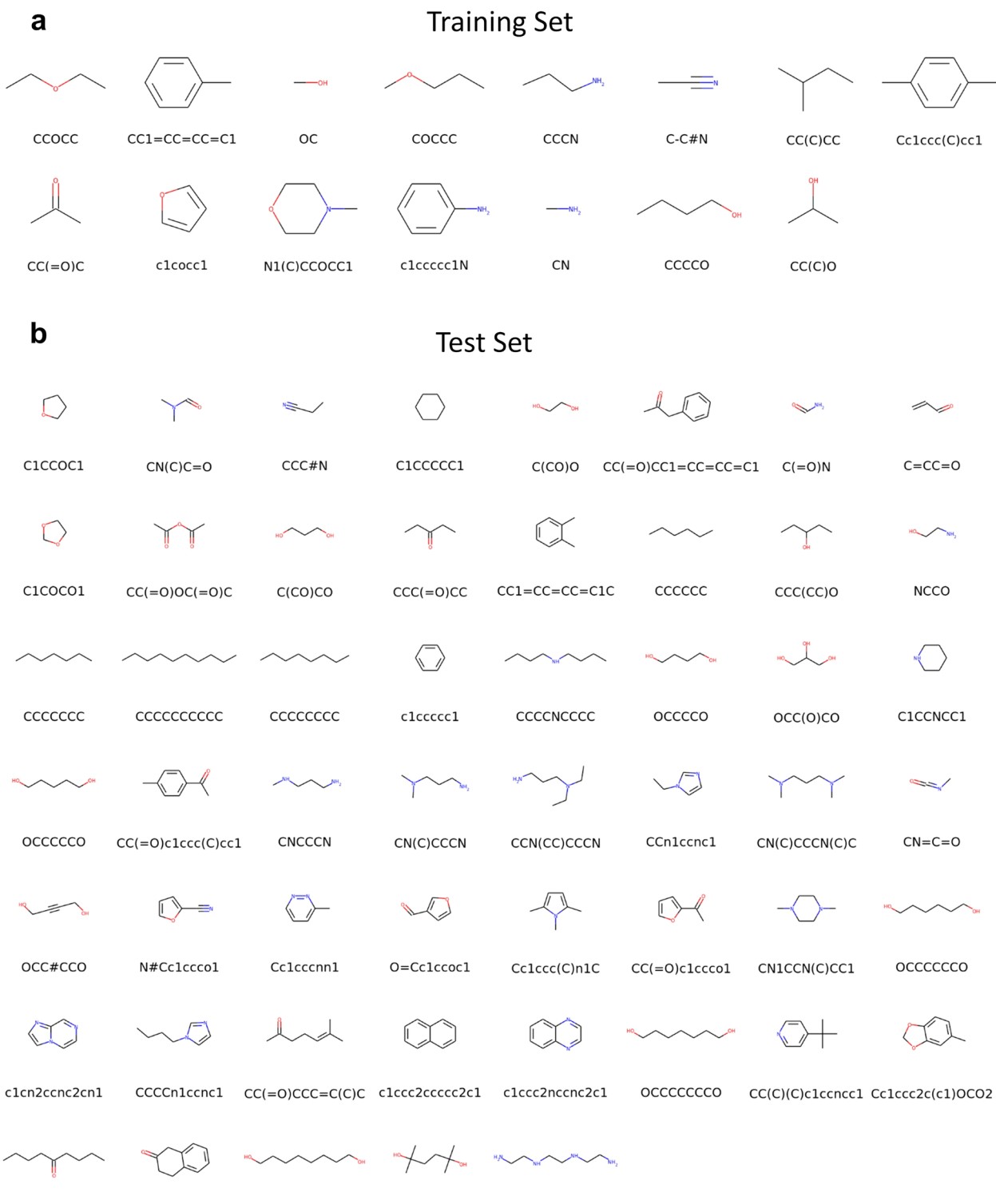

**Fig. 8 Molecules used in this study.** Molecules in **a** were used to train new LJ models, whereas the molecules in **b** were used to test the new parameters. The SMILES string for each molecule is given under the chemical structure.

$T = 298$ K were run for each molecule, to enable the calculation of liquid state properties and of the HOV. ForceBalance was also used to set up single-point simulations with baseline SMIRNOFF parameters, as well as with optimized parameters for the test set molecules after optimization on the training set. HOV were calculated as the gas/liquid difference between the mean potential energy per molecule plus the pressure–volume term RT. Liquid state densities were calculated from the mean volumes of the liquid state NPT simulations. Dielectric constants of liquids were calculated from the fluctuations of the simulations box's dipole moments, as implemented in ForceBalance[41].

For all simulations, the bonded FF terms were drawn from SMIRNOFF v1.0.7, and covalent bonds to hydrogen atoms were constrained to their equilibrium lengths with CCMA and SETTLE(water)[81,82]. Single-molecule gas-phase simulations were run for 25 ns (5 ns equilibration, 20 ns production) with a timestep of 1 fs using a Langevin integrator with a collision frequency of 1 ps$^{-1}$ and, infinite distance cutoffs and without periodic boundary conditions. Liquid-phase calculations, with 700 molecules in the box, were run for 1.2 ns (0.2 ns equilibration, 1 ns production), with a Langevin integrator timestep of 1 fs and a collision frequency of 1 ps$^{-1}$. The pressure was maintained at 1 atm with a Monte

Carlo barostat with a move attempt interval of 25 timesteps[83]. For the liquids, long-ranged electrostatics were included via Particle Mesh Ewald summation with a cutoff of 8.5 Å. A long-range dispersion correction was applied.

HFE were computed alchemically with the YANK program[34]. The temperature and pressure were set to 298 K and 1 atm, respectively, and the TIP3P water model was used. We used 5 lambda values in gas-phase and 20 lambda windows (5 windows for the electrostatics and 15 for the steric interactions) for the solution. The calculations used Hamiltonian replica exchange over 1000 iterations consisting each of 500 timesteps of 2 femtoseconds each. Analysis was done using Yank's standard analysis framework, which is based on multistate Bennett acceptance ratios[84].

All reference data and example input files to conduct this study, as well as the optimized force fields and charge parameters are available on GitHub (https://github.com/MSchauperl/RESP2). Additionally, a python library with a tool to parameterize molecules with RESP2 charges, including examples can be downloaded.

## Data availability
The data supporting the findings of this study are available within the article and its Supplementary Information files. All other relevant source data are available on https://github.com/MSchauperl/RESP2, https://doi.org/10.5281/zenodo.3593762, or from the corresponding authors upon reasonable request.

## Code availability
The code used to generate the results is available on https://github.com/MSchauperl/RESP2 and https://github.com/lpwgroup/respyte.

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

## Acknowledgements

M.S. acknowledges support of the Austrian Science Fund (FWF): Erwin Schrödinger fellowship J-4150. P.S.N. acknowledges the support of NASA Minority University Research and Education Project (MUREP) Institutional Research Opportunity grant NNX15AQ06A. L.-P.W. and H.J. acknowledge support from the ACS Petroleum Research Fund, award #58158-DNI6. M.K.G. acknowledges funding from National Institute of General Medical Sciences (GM61300). The contents of this paper are solely the responsibility of the authors and do not necessarily represent the official views of the funders.

## Author contributions

M.S., P.S.N., C.I.B., D.L.M., and M.K.G. conceived and designed the study. M.S. and H.J. performed the study. M.S., P.S.N., L.-P.W., C.I.B., and M.K.G. analyzed the data. M.S., oL.-P.W., and M.K.G. contributed reagents/materials/computational resources. M.S., P.S.N., D.L.M., and M.K.G. wrote the paper. All authors have reviewed the manuscript and have given approval to the final version.

## Competing interests

The authors declare the following competing interest(s): M.K.G. has an equity interest in and is a cofounder and scientific advisor of VeraChem LLC. All other authors declare no competing interests.
