## [Peer Review File · Communications Chemistry]

Reviewers' comments:

Reviewer #1 (Remarks to the Author):

The manuscript presents a protocol, named RESP2, to compute atomic partial charges for small-molecule force field simulations, where RESP stands for Restrained ElectroStatic Potential [fitting]. The RESP2 methodology uses the same restrained least-squares algorithm as the original RESP scheme to fit atomic partial charges to the electrostatic potential on a set of grid points surrounding a molecule. In RESP2, the least-squares algorithm is employed in revised and more modern workflow, comprising also conformer generation, local geometry optimization, density-functional theory (DFT) calculations with and without continuum solvent model, and a weighted average between the charges from gas phase and continuum DFT calculations. The RESP2 scheme is applied to various test cases, most importantly an assessment of how well molecular dynamics simulations using RESP2 partial charges can predict various experimental properties of organic molecules (in the condensed phase). An overall improvement is observed, relative to the original RESP method, when also (a reduced set of) Lennard-Jones (LJ) parameters is optimized with the partial charges. In particular, RESP2 charges from a weighted average of 40% gas phase and 60% continuum solvent DFT calculations, exhibit an optimal performance.

Overall the manuscript is very readable and leaves little room for uncertainty regarding the research methodology, the results and the conclusions. The availability of all data in an online GitHub repository is much appreciated. It would be useful to deposit the data also in another repository, e.g. Zenodo, with some guarantees on long-term availability.

A particular strength of this work is that it offers a pragmatic solution to the common issue of how to set up a fair non-polarizable force field for any small molecule (containing elements C, H, O and N) in a condensed phase simulation, despite all the approximations inherent to such models. The proposed workflow is easily reproducible and can be carried out with open source software, making it in principle accessible to a broad audience.

My main concern is that the title covers only one aspect of the paper, namely the method to compute the partial charges, while the work flow as a whole, including the fitted LJ parameters, makes this work most appealing. The RESP2 method is very similar to IPolQ-Mod, which the authors clearly mention, showing that the computation of the charges is not the ground-breaking innovation of this work. Essentially, IPolQ-Mod is refined, with a particular choice of DFT method and a tuning of the weighted average. In contrast, the effective and simplified parameterization of the LJ parameters is a more surprising and innovative aspect. Also the dependency of the optimal LJ parameters on the partial charges is an important new insight, supported by clear evidence. To frame it more negatively, this work does not convince that RESP2 charges without the refined LJ parameters offer a significant improvement over the state of the art, e.g. compared to IPolQ-Mod. This is not meant to downplay the importance of the manuscript, which is clearly of major importance. I merely recommend to shift the focus towards the force field parameterization protocol as a whole.

In addition to the main comment, I also have a few technical remarks:

- It is reassuring to see that the HF/6-31G* approach is dismissed and replaced by a more physically motivated approach to describe the polarization of the partial charges due to the condensed phase environment. However, using just a single value of 80 as relative dielectric constant to model the condensed phase seems inappropriate for the large variety of bulk phases considered in this work, e.g. when computing the densities of various less polar liquids with NpT molecular dynamics simulations. It would be helpful to explain in the manuscript why this high dielectric constant still produces reasonable RESP2 charges for all liquids.

- A minor comment on the paragraph starting at line 484 is that also the dielectric constant can

have an important contribution from electronic polarization, which a fixed-charge model does not address. Just like the heat of vaporization and the hydration free energy, it may go beyond the scope of a fixed-charge model to predict dielectric constants with high accuracy. The absence of electronic polarization in a fixed-charge model could also explain why the simulated dielectric constants in figure S1E generally underestimate the experimental reference values.

- Which RESP settings were used, e.g. what is the magnitude of the restraint coefficient? Which constraints were applied during the charge fitting, e.g. symmetry?

- The protocol prescribes a local optimization of the molecular geometries generated by Omega, with a gas phase DFT calculation [PW6B95/cc-pV(D+d)Z]. Could this bias the charge fitting to gas-phase geometries, e.g. with internal hydrogen bonds?

Finally, some minor comments:

- Could the authors include numerical values of the optimized Lennard-Jones parameters in the Supporting Information?

- Some of the mathematical symbols in the manuscript seem to be incorrect, e.g. δ instead of ϵ on the last line of page 8 and also on page 14. It would also be helpful to use either σ or $r_{\text{min-half}}$ systematically.

Reviewer #2 (Remarks to the Author):

The authors present an update of the popular RESP model for deriving atomic partial charges for use in MM force fields (named RESP2).

The key difference compared to RESP1 is that, rather than using the HF/6-31G* QM method and hoping for fortuitous overpolarization suitable for the condensed phase, the authors propose a weighted average between gas phase and aqueous charges. They also advocate reoptimization of the Lennard-Jones parameters with each new charge model, a consideration which is all too often overlooked.

With today's computing resources, movement away from the HF/6-31G* charge assignment model cannot come soon enough. If shown to be more accurate than RESP1, users will put up with the extra computational expense. As such, I believe that this study will be of broad interest across the molecular modelling community and should be published with the following minor considerations.

- The method does not seem to be very significantly different to the published iPolQ-Mod approach, differing only by the fitting parameter δ . In fact, the authors show that the accuracy of using $\delta=0.5$ is indistinguishable from using their proposed $\delta=0.6$. Since $\delta=0.5$ has some physical justification is it not tempting to keep it fixed at 0.5? The accuracy of RESP2 in eg Fig 5 also does not seem all that high. Perhaps most interesting (as the authors point out) is the relative insensitivity of the accuracy to δ when LJ parameters are optimized. Are we reaching the limit of this functional form of the force field, or can other force fields eg OPLS3 perform better on

test sets like this one? Some more accuracy comparisons with the literature would be helpful here.

- The authors make some methodological choices that are not explicitly discussed. Presumably, the same underlying function and parameters from the RESP1 method are used to restrain charges towards zero to avoid ill-defined charges on buried atoms? This seems a little counterintuitive now that we want the charges to polarize in response to the implicit solvent. Nevertheless, Fig 7 seems to suggest that polarization is working as expected in RESP2. In any case, these charge restraint methods should at least be summarized since the authors are proposing an important update to RESP.

- Similarly, the authors are using the CPCM implicit solvent model (the methods state both PCM and CPCM on page 5?), which is itself heavily parameterized and not a unique solution to finding the aqueous phase ESP. Would using e.g. IPCM (using an isodensity surface to define the solvent cavity) be consistent with the RESP2 method? I'd be interested in a plot in Fig 8 showing correlations between charges derived using a different implicit solvent model. They may well differ by as much as RESP2 differs from RESP1.

- The authors correctly state that charge and Lennard-Jones parameters are interdependent, and should be optimized together. What about dihedral parameters? Some of the test set molecules are very flexible, and drastic changes to the sigma and epsilon parameters will change the torsional profiles. The authors could check the effects on the liquid properties, eg by refitting the torsional parameters for one of the molecules.

Minor comments:

- I found the abstract a bit long for a communication, but this may be personal preference.

- page 5. the dielectric constant of the aqueous phase is described as $\epsilon=80$, in other places it is $\epsilon=78.39$.

- page 8, L222 (and page 14). The parameters to be fit should be r_{min} and ϵ . The second character is unclear in my version of the pdf.

- page 9, L227. Should the prior widths have units?

- Figure 3 (center). I would expect the ESP error to have units of q/r in atomic units? In any case, I don't have a good sense of how accurate or otherwise an error of $1 e^2/(a_0^2)$ is. Would it be better to consider the electrostatic potential energy of a unit test charge, and convert to energy units eg kcal/mol?

Reviewer #3 (Remarks to the Author):

The paper by Schauerl et al. reports on an extension of the well established RESP method to fit atomic point charges for organic molecules. In the RESP method a fit to the electrostatic potential is performed with a constraint to keep charges of similar type (symmetry, atomtype) equal. A HF density is used which leads to overpolarization in order to mimic the effect of a polar solvent. In the current work an extension RESP2 is proposed, where a more accurate hybrid meta-GGA is used together with an extended basis set. Instead of relying on the fortuitous overpolarization at the HF level, two calculations in the gas phase and in a continuum solvent are performed and the charges are linearly interpolated. With this to some extent obvious extension of the method, the authors do not observe a substantial improvement in various molecular and bulk parameter. However, with a joint re-parameterization of the LJ vdW interaction an improvement with respect to the original baseline parameterization is found.

The paper is well written and the methods are state of the art. It contains a lot of detailed information of relevance for the practitioners of MD simulation of organic molecular systems. On the other hand, the presented methodology is not very innovative, since it relies on a simple combination of existing and well established methods and the general idea is rather straight forward.

An aspect, which is in my opinion missing in the paper is a short discussion of other methods to determine charges for classical force field methods, which have been developed over the years, since the RESP method from is from 1993. First of all, a lot of effort has been invested in charge determination methods in context of materials science and in particular in the context of porous materials. Here REPEAT charges have been established for periodic systems. Other relevant approaches are the DDEC charges by Manz et. al., addressing the problem of buried atoms (a problem not even mentioned in this paper!) by combining ESP fits with Hirshfeld-like methods. Another very promising method is in my opinion the MBIS (minimal basis iterative stockholder) charges by Verstraelen et al.. Also a number of groups have worked on machine learning methods to predict atomic charges. This is a by no means complete list of alternatives.

Minor points:

Some of the graphics in the ESI (especially S1) contain very small font text (some legends are hard to read).

Reviewer #1 (Remarks to the Author):

The manuscript presents a protocol, named RESP2, to compute atomic partial charges for small-molecule force field simulations, where RESP stands for Restrained ElectroStatic Potential [fitting]. The RESP2 methodology uses the same restrained least-squares algorithm as the original RESP scheme to fit atomic partial charges to the electrostatic potential on a set of grid points surrounding a molecule. In RESP2, the least-squares algorithm is employed in a revised and more modern workflow, comprising also conformer generation, local geometry optimization, density-functional theory (DFT) calculations with and without continuum solvent model, and a weighted average between the charges from gas phase and continuum DFT calculations. The RESP2 scheme is applied to various test cases, most importantly an assessment of how well molecular dynamics simulations using RESP2 partial charges can predict various experimental properties of organic molecules (in the condensed phase). An overall improvement is observed, relative to the original RESP method, when also (a reduced set of) Lennard-Jones (LJ) parameters is optimized with the partial charges. In particular, RESP2 charges from a weighted average of 40% gas phase and 60% continuum solvent DFT calculations, exhibit an optimal performance.

We thank the reviewer for this positive overall assessment.

Overall the manuscript is very readable and leaves little room for uncertainty regarding the research methodology, the results and the conclusions. The availability of all data in an online GitHub repository is much appreciated. It would be useful to deposit the data also in another repository, e.g. Zenodo, with some guarantees on long-term availability.

We agree and have made the data available on Zenodo: <https://zenodo.org/badge/latestdoi/196089790>

A particular strength of this work is that it offers a pragmatic solution to the common issue of how to set up a fair non-polarizable force field for any small molecule (containing elements C, H, O and N) in a condensed phase simulation, despite all the approximations inherent to such models. The proposed workflow is easily reproducible and can be carried out with open source software, making it in principle accessible to a broad audience.

Again, thanks to the reviewer.

My main concern is that the title covers only one aspect of the paper, namely the method to compute the partial charges, while the work flow as a whole, including the fitted LJ parameters, makes this work most appealing. The RESP2 method is very similar to IPolQ-Mod, which the authors clearly mention, showing that the computation of the charges is not the ground-breaking innovation of this work. Essentially, IPolQ-Mod is refined, with a particular choice of DFT method and a tuning of the weighted average. In contrast, the effective and simplified parameterization of the LJ parameters is a more surprising and innovative aspect. Also the dependency of the optimal LJ parameters on the partial charges is an important new insight, supported by clear evidence. To frame it more negatively, this work does not convince that RESP2 charges without the refined LJ parameters offer a significant improvement over the state of the art, e.g. compared to IPolQ-Mod. This is not meant to downplay the importance of the manuscript, which is clearly of major importance. I merely recommend to shift the focus towards the force field parameterization protocol as a whole.

We appreciate and agree with the reviewer's suggestion. We have changed the title, abstract, and discussion to shift the focus to the entire approach, both the RESP2 charges and the fitting of the LJ parameters.

In addition to the main comment, I also have a few technical remarks:

- It is reassuring to see that the HF/6-31G* approach is dismissed and replaced by a more physically motivated approach to describe the polarization of the partial charges due to the condensed phase environment. However, using just a single value of 80 as relative dielectric constant to model the condensed phase seems inappropriate for the large variety of bulk phases considered in this work, e.g. when computing the densities of various less polar liquids with NpT molecular dynamics simulations. It would be helpful to explain in the manuscript why this high dielectric constant still produces reasonable RESP2 charges for all liquids.

This is an interesting question. Our thought is that the errors from using the dielectric constant of water for all of these organic compounds may be less than perhaps anticipated because apolar molecules generate only a weak reaction field no matter what the surrounding dielectric constant, so the precise value may not matter much; while polar molecules, for which the surrounding dielectric constant matters more, have a dielectric constant closer to water.

- A minor comment on the paragraph starting at line 484 is that also the dielectric constant can have an important contribution from electronic polarization, which a fixed-charge model does not address. Just like the heat of vaporization and the hydration free energy, it may go beyond the scope of a fixed-charge model to predict dielectric constants with high accuracy. The absence of electronic polarization in a fixed-charge model could also explain why the simulated dielectric constants in figure S1E generally underestimate the experimental reference values.

We thank the reviewer for pointing this out and now discuss this in the revised manuscript.

- Which RESP settings were used, e.g. what is the magnitude of the restraint coefficient? Which constraints were applied during the charge fitting, e.g. symmetry?

We agree with the reviewers that this information is important. We have added a paragraph to the Methods section specifying the settings. Additionally, example input files with the used settings are available on GitHub.

- The protocol prescribes a local optimization of the molecular geometries generated by Omega, with a gas phase DFT calculation [PW6B95/cc-pV(D+d)Z]. Could this bias the charge fitting to gas-phase geometries, e.g. with internal hydrogen bonds?

We agree with the reviewer that the selected geometries could be important, so we inspected all our molecule's geometries. We did not find an unusual amount of intramolecular hydrogen bonds or compact optimized structures (increased amount of van der Waals contacts), so we do not think this is

in fact an issue here. We have added text to this effect in the Methods section.

Finally, some minor comments:

- Could the authors include numerical values of the optimized Lennard-Jones parameters in the Supporting Information?

We have included the LJ parameters for RESP2 ($\delta=0.5$, $\delta=0.6$) and RESP1 in the supporting information. Additionally, the values are on GitHub in an offxml file.

- Some of the mathematical symbols in the manuscript seem to be incorrect, e.g. δ instead of ϵ on the last line of page 8 and also on page 14. It would also be helpful to use either σ or $r_{\text{min-half}}$ systematically.

We thank the reviewer for his or her careful reading of our manuscript; we have corrected these symbols.

Reviewer #2 (Remarks to the Author):

The authors present an update of the popular RESP model for deriving atomic partial charges for use in MM force fields (named RESP2).

The key difference compared to RESP1 is that, rather than using the HF/6-31G* QM method and hoping for fortuitous overpolarization suitable for the condensed phase, the authors propose a weighted average between gas phase and aqueous charges. They also advocate reoptimization of the Lennard-Jones parameters with each new charge model, a consideration which is all too often overlooked.

With today's computing resources, movement away from the HF/6-31G* charge assignment model cannot come soon enough. If shown to be more accurate than RESP1, users will put up with the extra computational expense. As such, I believe that this study will be of broad interest across the molecular modelling community and should be published with the following minor considerations.

We thank the reviewer for this positive assessment.

- The method does not seem to be very significantly different to the published iPolQ-Mod approach, differing only by the fitting parameter delta. In fact, the authors show that the accuracy of using delta=0.5 is indistinguishable from using their proposed delta=0.6. Since delta=0.5 has some physical justification is it not tempting to keep it fixed at 0.5?

As also mentioned by the reviewer (below), the PCM model contains empirical parameters that can influence the degree of self-polarization; in addition, there are different flavors of PCM. Therefore, it is possible that different values of δ would be optimal depending on the details of the PCM method used. In addition, there is evidence that PCM models underpolarize molecules, in comparison to explicit solvent (personal correspondence with Dr. Julia Rice). The adjustment of the mixing parameter δ allows

us to compensate empirically for all of these variations, and potentially also for fundamental limitations of the simple functional form of the force field.

The accuracy of RESP2 in eg Fig 5 also does not seem all that high. Perhaps most interesting (as the authors point out) is the relative insensitivity of the accuracy to delta when LJ parameters are optimized. Are we reaching the limit of this functional form of the force field, or can other force fields eg OPLS3 perform better on test sets like this one? Some more accuracy comparisons with the literature would be helpful here.

Our impression is that there is still value in optimizing parameters for this functional form, though we are also very much interested in moving to more sophisticated functional forms, following the important work of others in the field. It is also worth noting that the current study optimizes only non-bonded terms; additional accuracy may be gained when the torsions are refitted with the new LJ parameters. The revised Discussion section makes these points and also touches on other force fields, such as OPLS3..

- The authors make some methodological choices that are not explicitly discussed. Presumably, the same underlying function and parameters from the RESP1 method are used to restrain charges towards zero to avoid ill-defined charges on buried atoms? This seems a little counterintuitive now that we want the charges to polarize in response to the implicit solvent. Nevertheless, Fig 7 seems to suggest that polarization is working as expected in RESP2. In any case, these charge restraint methods should at least be summarized since the authors are proposing an important update to RESP.

We thank the reviewer for pointing out that this important information is missing. As now detailed in the revised Methods section, we use the same two stage fitting process as RESP1. We believe it is still important to restrain charges gently toward zero, as the presence of the implicit solvent does not resolve the basic problem that the charges of buried atoms are only weakly defined by fitting to the ESP and thus risk taking on nonphysical values.

- Similarly, the authors are using the CPCM implicit solvent model (the methods state both PCM and CPCM on page 5?), which is itself heavily parameterized and not a unique solution to finding the aqueous phase ESP. Would using e.g. IPCM (using an isodensity surface to define the solvent cavity) be consistent with the RESP2 method? I'd be interested in a plot in Fig 8 showing correlations between charges derived using a different implicit solvent model. They may well differ by as much as RESP2 differs from RESP1.

A systematic change (e.g. systematically more polar) in the PCM model should only lead to a shift in the mixing parameter δ and therefore should not change our results (except for the mixing parameter). The influence of unsystematic changes is hard to grasp. Especially as we cannot judge which method leads to the more correct charges. We examined the potential consequences of switching from CPCM to IPCM, both with default values in psi4, by using both to compute RESP2 partial charges for 1,6 -Hexanediol, using $\delta=0.5$. The two sets of charges correlate with $R^2=0.9999$, linear regression slope of 1.0096 and intercept of $-5e-20$. Thus, changing to IPCM would likely have little effect on the present study.

Atom	RESP1	RESP2 ($\delta=0.5$)	RESP2 ($\delta=0.5, \text{IPCM}$)
C1	0.020	0.031	0.033

C2	0.020	0.031	0.033
C3	0.024	0.018	0.014
C4	0.024	0.018	0.014
C5	0.169	0.143	0.143
C6	0.169	0.143	0.143
	-		
O1	0.600	-0.588	-0.593
	-		
O2	0.600	-0.588	-0.593
	-		
H1	0.008	-0.009	-0.010
	-		
H2	0.008	-0.009	-0.010
	-		
H3	0.008	-0.009	-0.010
	-		
H4	0.008	-0.009	-0.010
H5	0.003	0.005	0.006
H6	0.003	0.005	0.006
H7	0.003	0.005	0.006
H8	0.003	0.005	0.006
H9	0.013	0.022	0.023
H10	0.013	0.022	0.023
H11	0.013	0.022	0.023
H12	0.013	0.022	0.023
H13	0.372	0.360	0.365
H14	0.372	0.360	0.365

- The authors correctly state that charge and Lennard-Jones parameters are interdependent, and should be optimized together. What about dihedral parameters? Some of the test set molecules are very flexible, and drastic changes to the sigma and epsilon parameters will change the torsional profiles. The authors could check the effects on the liquid properties, eg by refitting the torsional parameters for one of the molecules.

We agree that a full force field optimization would require refitting of the torsions, but this would dramatically expand the scope of our study. We hope the present approach will find application within the broader Open Force Field initiative (openforcefield.org).

Minor comments:

- I found the abstract a bit long for a communication, but this may be personal preference.

We have rephrased and shortened the abstract.

- page 5. the dielectric constant of the aqueous phase is described as $\epsilon=80$, in other places it is $\epsilon=78.39$.

We thank the reviewer for his or her careful reading of our manuscript. We have changed all values to the in this study used epsilon value of 78.39.

- page 8, L222 (and page 14). The parameters to be fit should be r_{min} and epsilon. The second character is unclear in my version of the pdf.

Yes, something went wrong in converting from Word to pdf; this should now be corrected.

- page 9, L227. Should the prior widths have units?

We added units to the prior widths.

- Figure 3 (center). I would expect the ESP error to have units of q/r in atomic units? In any case, I don't have a good sense of how accurate or otherwise an error of $1 e^2/(a_0^2)$ is. Would it be better to consider the electrostatic potential energy of a unit test charge, and convert to energy units eg kcal/mol?

We thank the reviewer for their careful reading. There was indeed a wrong unit on the y-axis.

We also followed the reviewer's suggestion about converting units to kJ / mol / elementary charge.

Reviewer #3 (Remarks to the Author):

The paper by Schauperl et al. reports on an extension of the well established RESP method to fit atomic point charges for organic molecules. In the REST method a fit to the electrostatic potential is performed with a constraint to keep charges of similar type (symmetry, atomtype) equal. A HF density is used which leads to overpolarization in order to mimic the effect of a polar solvent. In the current work an extension RESP2 is proposed, where a more accurate hybrid meta-GGA is used together with an extended basis set. Instead of relying on the fortuitous overpolarization at the HF level, two calculations in the gas phase and in a continuum solvent are performed and the charges are linearly interpolated. With this to some extent obvious extension of the method, the authors do not observe a substantial improvement in various molecular and bulk parameter. However, with a joint re-parameterization of the LJ vdW interaction an improvement with respect to the original baseline parameterization is found.

The paper is well written and the methods are state of the art. It contains a lot of detailed information of relevance for the practitioners of MD simulation of organic molecular systems. On the other hand, the presented methodology is not very innovative, since it relies on a simple combination of existing a well established methods and the general idea is rather straight forward.

We thank the reviewer for this thoughtful assessment. Although the idea itself might be straightforward, we would argue that our approach is an advance. In addition to its more accurate description of charges, we would highlight the concept of optimizing the overall polarity of the charge set (i.e., the value of δ) along with the LJ parameters.

An aspect, which is in my opinion missing in the paper is a short discussion of other methods to determine charges for classical force field methods, which have been developed over the years, since the RESP method from is from 1993. First of all, a lot of effort has been invested in charge determination methods in context of materials science and in particular in the context of porous materials. Here REPEAT charges haven been established for periodic systems. Other relevant approaches are the DDEC

charges be Manz et. al., addressing the problem of buried atoms (a problem not even mentioned in this paper!) by combining ESP fits with Hirshfeld-like methods. Another very promising method is in my opinion the MBIS (minimal basis iterative stockholder) charges by Verstraelen et al.. Also a number of groups have worked on machine learning methods to predict atomic charges. This is a by no means complete list of alternatives.

We have added citations to these methods in the revised Introduction, but have not added a broader discussion as this would cause the manuscript to exceed the journal's 1000 word limit. However, we have included some text about buried atoms in the revised Methods section.

Minor points:

Some of the graphics in the ESI (especially S1) contain very small font text (some legends are hard to read).

We tried to change the font size in Supplementary Figure 1. However, when we increase the font-size, the labels overlap more, and thus reduced readability, so we have elected to leave the font unchanged.

REVIEWERS' COMMENTS:

Reviewer #1 (Remarks to the Author):

The revised manuscript addresses all my concerns and I have no further comments.

Reviewer #2 (Remarks to the Author):

The authors have fully addressed my previous comments, and so I support publication of the manuscript in its current form.

- Daniel J. Cole (Newcastle University)

Reviewer #3 (Remarks to the Author):

The revised paper is now publishable. Also in the light of the discussion with the other reviewers comments (and the change in title and focus) the paper is now much clearer to me and I can see the point. The authors have addressed my points but - in my opinion - also the criticism of the other reviewers and the manuscript has gained a lot in strength and clarity in the review.